# Policies for type 2 diabetes and non-communicable disease management during the COVID-19 pandemic in Kenya and Tanzania: a desk review and views of decision-makers

Shukri F Mohamed ,[1] Lyagamula Kisia ,[1] Irene Mashiashi,[2] Frances Mair ,[3] Jennifer P Wisdom,[4] Christopher Bunn,[5] Cindy Gray ,[5] Peter M Kibe,[1] Richard E Sanya ,[1] Caroline H Karugu ,[1] Sally M Mtenga,[2] Peter Binyaruka ,[2] Gershim Asiki[1]

**Correspondence to**
Shukri F Mohamed;
smohamed@aphrc.org

## ABSTRACT

**Background** The COVID-19 pandemic caused disruptions in care that adversely affected the management of non-communicable diseases (NCDs) globally. Countries have responded in various ways to support people with NCDs during the pandemic. This study aimed to identify policy gaps, if any, in the management of NCDs, particularly diabetes, during COVID-19 in Kenya and Tanzania to inform recommendations for priority actions for NCD management during any future similar crises.

**Methods** We undertook a desk review of pre-existing and newly developed national frameworks, policy models and guidelines for addressing NCDs including type 2 diabetes. This was followed by 13 key informant interviews with stakeholders involved in NCD decision-making: six in Kenya and seven in Tanzania. Thematic analysis was used to analyse the documents.

**Results** Seventeen guidance documents were identified (Kenya=10; Tanzania=7). These included pre-existing and/or updated policies/strategic plans, guidelines, a letter, a policy brief and a report. Neither country had comprehensive policies/guidelines to ensure continuity of NCD care before the COVID-19 pandemic. However, efforts were made to update pre-existing documents and several more were developed during the pandemic to guide NCD care. Some measures were put in place during the COVID-19 period to ensure continuity of care for patients with NCDs such as longer supply of medicines. Inadequate attention was given to monitoring and evaluation and implementation issues.

**Conclusion** Kenya and Tanzania developed and updated some policies/guidelines to include continuity of care in emergencies. However, there were gaps in the documents and between policy/guideline documents and practice. Health systems need to establish disaster preparedness plans that integrate attention to NCD care to enable them to better handle severe disruptions caused by emergencies such as pandemics. Such guidance needs to include contingency planning to enable adequate resources for NCD care and must also address evaluation of implementation effectiveness.

## STRENGTHS AND LIMITATIONS OF THIS STUDY

⇒ Triangulation of data sources from desk review and qualitative interviews.
⇒ Comprehensive approach to identify and extract data in a standardised format.
⇒ Engagement with key stakeholders in non-communicable disease decision-making to retrieve non-public documents.
⇒ Findings limited to documented and accessible information.
⇒ The absence of patient and healthcare provider perspectives may have hindered a comprehensive understanding of policy implementation, successes, challenges, to inform recommendations for improvement.

## INTRODUCTION

The rapid spread of COVID-19 within a short timeframe resulted in a global pandemic, with a current toll of over six million deaths worldwide as of 8 November 2023.[1] In Africa, the impact has been significant, with over 9 million confirmed cases and nearly 175 500 deaths reported in the 3 years since the initial detection of COVID-19.[1] While Africa's share of the pandemic's morbidity and mortality is comparatively lower than other regions, the region faced challenges due to its fragile healthcare system, existing comorbidities and socioeconomic factors.[2–4] The response to the pandemic varied among African countries. For example, Kenya implemented strict lockdown measures,[5] while Tanzania opted not to impose such measures.[6] Many countries faced challenges in adequately preparing for the multifaceted challenges posed by COVID-19, putting additional strain on already stretched resources.[7]

This necessitated a concerted effort for the development and implementation of measures to mitigate the pandemic's impact and strengthen healthcare systems to address both new and existing health challenges.[8]

The COVID-19 pandemic caused disruptions in care that adversely affected the management of non-communicable diseases (NCDs) globally. Government restrictions implemented in various countries to curb the spread of COVID-19 such as lockdown directives and mobility restrictions impacted management of NCDs by individuals and health systems.[9] A rapid assessment conducted by the WHO across 163 countries revealed that NCD health services, including rehabilitation services, hypertension management and diabetes management, were partially or completely disrupted by the pandemic.[10] These disruptions are likely to have ongoing negative impacts on the management of chronic diseases and progress towards achieving sustainable development goal targets. Studies reported that patients with NCDs such as hypertension and diabetes were more likely than those without these comorbidities to be affected by COVID-19 and suffer worse health outcomes if infected.[11 12]

Despite most countries prioritising services for four major NCDs (diabetes, hypertension, cancer and cardiovascular disease) during the COVID-19 pandemic,[10] maintaining routine care for NCDs is a major challenge due to barriers at various levels of healthcare systems. Patients with type 2 diabetes (T2D) may need continuous and regular blood glucose monitoring and optimal compliance with ongoing drug treatment.[13] Healthcare providers reported that diabetes was the most affected condition by the reduction in healthcare resources.[9] The WHO assessment found that 66% of countries surveyed included the continuity of NCD services in the national COVID-19 plans.[10]

Countries have used various strategies to support people with NCDs including diabetes during this pandemic,[14] primarily switching from routine care visits to virtual healthcare visits and communications.[9] Despite these adjustments in service delivery for NCDs, most global healthcare resources focused on the COVID-19 emergency response,[15] particularly in sub-Saharan Africa (SSA) where healthcare personnel and health resources are inadequate.[16] This resource reallocation disrupted care for patients with NCDs.[17]

In this review, we synthesise policy evidence regarding T2D management during COVID-19 and identified policy gaps, if any, in management of T2D and other NCDs to inform T2D management actions during COVID-19 in Kenya and Tanzania. Our research questions were as follows:

1. Are there pre-existing national frameworks, policy models (priorities, strategies and directions) and guidelines (instructions) for addressing T2D /NCDs? If so, did they cover pandemic/emergency situations? How have existing frameworks/models (includes formal framework/models and informal statements) been modified for pandemic/emergency situations (eg, COVID-19) or how have framework and policy models been created during COVID-19 (including those in development)?

2. What do the national frameworks, policy models or guidelines tend to include with regard to priority setting (access to care and medical supplies, resource allocation and monitoring and evaluation tools) for management of T2D /NCDs in pandemic/emergency situations?

3. What are the policy/guideline gaps for the management of T2D during the COVID-19 pandemic (eg, addressing existing disparities such as socioeconomic status and urban/rural status, addressing health service staff, financial resources for health management, implementation, and communication/dissemination), and what are the opportunities to improve the management of T2D/NCDs?

## METHODS

We conducted a systematic search of policy documents and guiding frameworks on NCDs with a particular focus on T2D in Kenya and Tanzania from October 2021 to March 2022. Our review included pre-existing national frameworks, policy models or guidelines, noting whether these had been modified to include pandemic/emergency situations, as well as new ones developed during the COVID-19 pandemic. Identified documents were assessed to ascertain policy/guideline gaps, if any, and opportunities to improve the management of T2D or other NCDs during periods where there are disruptions in care such as the COVID-19 pandemic. While the primary interest was T2D, we also included policy documents which broadly covered NCDs and reviewed their applicability to T2D.

This review was further supplemented by key informant interviews (KIIs)[18] conducted between March and August 2022. The aim of the interviews was to understand how the Kenyan and Tanzanian governments addressed the management of NCDs including T2D during COVID-19, with specific focus on the following: policy formulation during COVID-19, policy implementation during COVID-19 and gaps noted in the management of NCDs and T2D during COVID-19. Data from the review and the key informant's interview were analysed separately and then triangulated.

## DOCUMENT REVIEW
### Eligibility criteria

For inclusion in this review, documents had to meet four criteria: (1) policy frameworks, policy statements, government statements/pronouncements, publications, emergency preparedness plans, strategies, laws and operational guidelines for diabetes during COVID-19 (this also included policies under development); (2) specifically mentions NCDs or diabetes or pandemic/emergency or COVID-19; (3) printed in English/Kiswahili and (4) references Kenya or Tanzania. Documents were excluded from this review if they did not include information on

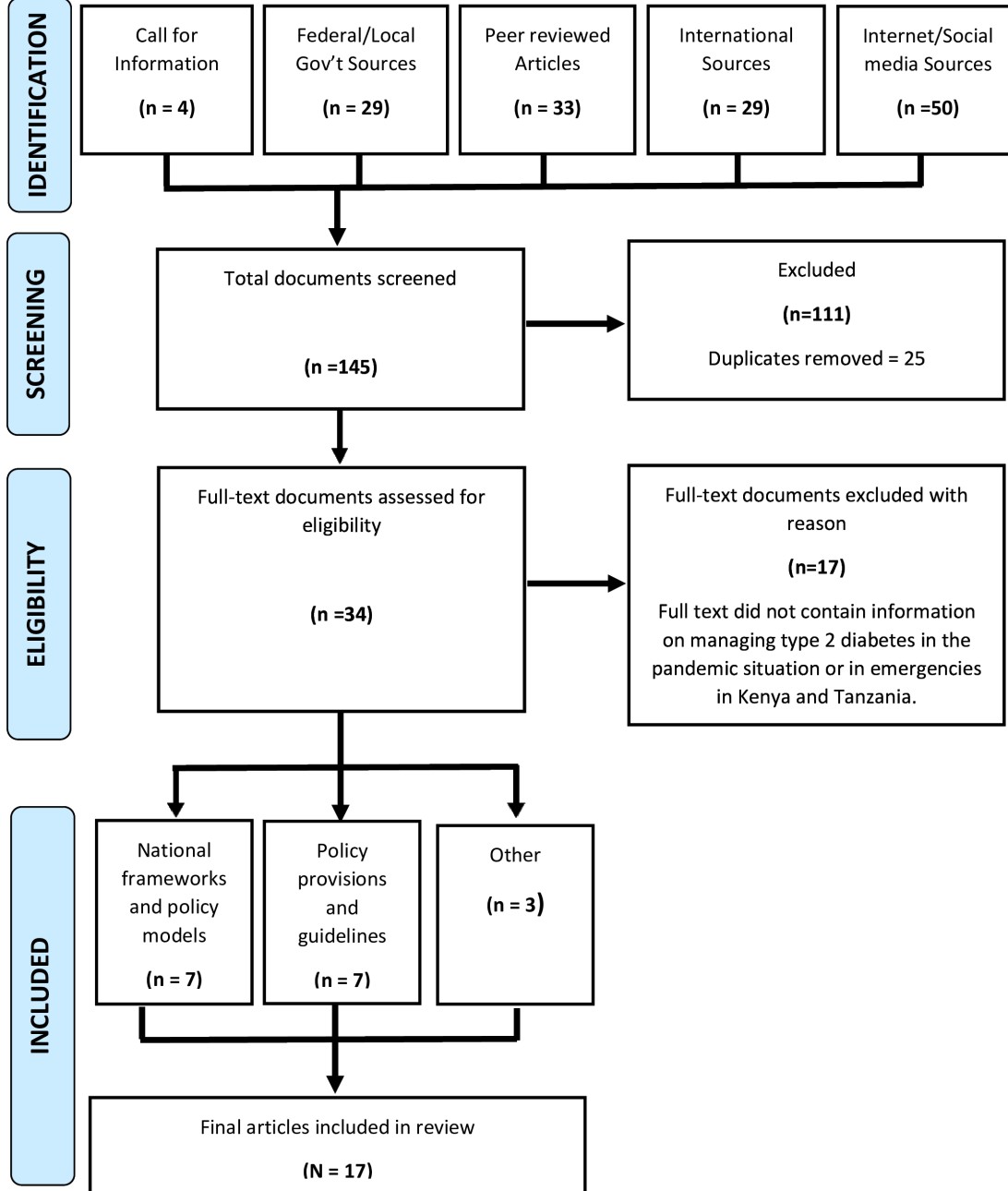

**Figure 1** Search strategy.

managing T2D in a pandemic/emergency situation and if the documents did not reference Kenya or Tanzania.

### Search strategy

A team of seven researchers (Kenya=4; Tanzania=3) with backgrounds in medicine, public health and social sciences conducted the searches and reviewed the documents. All received training on conducting desk reviews. The team coordinator assigned each researcher a different database to search for relevant documentation.

We searched using key text words such as 'Diabetes' and 'COVID-19', and 'Healthcare' and 'Policies' and 'Kenya' or 'Tanzania'. Online supplemental appendix A describes the specific terms used in the search. Using these terms, researchers searched scientific databases (PubMed/

Medline, Embase and Web of Science), Ministry of Health (MOH) websites, local/international resources such as the WHO and government social media such as the MOH Twitter and Instagram accounts, supplemented with Google searches. We also contacted existing networks and experts in the NCD field through a call for information which was sent via email and WhatsApp groups to ask for any other relevant policy documents. Online supplemental appendix B provides a detailed explanation of the search strategy for each data source of information.

All retrieved documents were catalogued in an online excel database which facilitated a standardised approach to data extraction from each document. The document characterisation included a unique document number

for tracking the title of the document, source of the document (eg, social media and MOH) and basic information about the document (eg, year document was produced). The team held weekly 'Coding clinics' to ensure consistency in coding of the included documents.

## KEY INFORMANT INTERVIEWS (KII)
### Design and participants
The results from the document review were used to inform development of the interview guides for the KIIs. In both Kenya and Tanzania, key informants (KIs) were purposively selected based on their experiences and participation in NCD-related policy formulation and implementation processes.

### Data collection procedures
A semistructured interview guide (online supplemental appendix C) was used to explore and understand: (a) what policies/guidelines/frameworks had been developed during COVID-19; (b) who and which sectors had been involved in the policy development and implementation; (c) challenges in the formulation and implementation; (d) gaps that still exist and (e) recommendations.

Interviews were conducted by two experienced qualitative researchers who were familiar with the NCDs/T2D policy documents. Interviews were conducted in English in Kenya and in Kiswahili in Tanzania. Participants chose between in-person or remote interviews to ensure convenience and mitigate risk. The interviews lasted approximately 45 min. All interviews were audio-recorded and complemented by handwritten notes.

### Data management and analysis
We used content analysis[19] to identify the nature of included documents, the content of the documents, what the documents included regarding priority setting (access to care and medical supplies, resource allocation and monitoring and evaluation tools) and the gaps and opportunities within each document. The information from the desk review was triangulated with the key informants' information to ensure a more in-depth understanding of the synthesised documents and to provide further insights into policy recommendations for NCD management for future emergency situations. The transcripts were coded manually in Word. Two independent researchers, experienced in qualitative data analysis, double-coded two transcripts in each country to ensure coding consistency. We obtained a per cent agreement of 82%. Disagreements were resolved by rereading the transcripts, discussion and a consensus was reached. In Tanzania, on request from the Tanzania NCD Alliance, a further narrative synthesis of the findings was shared with an NCD expert to identify if there was any important information that was left out, misinterpreted, or missed during the search.

### Patient and public involvement
Patients or the public were not involved in the design, or conduct, or reporting or dissemination of the research.

Results of the study were presented to study participants and stakeholders.

## RESULTS
Figure 1 describes the search results. After excluding duplicates and records that were not relevant based on title and abstract screening, we identified 34 policy/guidance documents for full-text review. After full-text review, 17 documents were further excluded because they did not contain information on managing T2D in a pandemic or emergency situation, leaving 17 documents that were included in this review.

Among the included documents, 10 were from Kenya and seven were from Tanzania. In Kenya, we found three interim guidelines, two updated guidelines and a letter from the Director General of Health. In Tanzania, two modified guidelines, one policy brief and one report were included in the review. Each country had one pre-existing national health policy, one pre-existing NCD strategic plan, and an updated NCD strategic plan.

In total, 13 KIIs (Tanzania=7; Kenya=6) were conducted with policy actors between March and August 2022. All participants were in the health sector and had different backgrounds in medicine, epidemiology, strategic management and nutrition. Most of the participants served in various capacities within the MOH. Table 1 provides the characteristics of the KIs interviewed.

### Research question 1: what national frameworks, policy models and guidelines for addressing type 2 diabetes/NCDs pre-existed and which ones were modified or developed during the COVID-19 pandemic?
#### Pre-existing documents
There were five documents in Kenya and Tanzania that addressed NCDs or emergency situations prior to the COVID-19 pandemic but none specifically provided guidance on NCD preparedness in an emergency situation. Table 2 and online supplemental appendix D provides a brief description of the included documents. The pre-existing documents include the following: the Tanzania

**Table 1** Characteristics of key informants

|  | Kenya | Tanzania | Total |
|---|---|---|---|
| **Category** |  |  |  |
| Government | 6 | 4 | 10 |
| Bilateral organisations/NGOs | 0 | 3 | 3 |
| **Years served in that capacity** |  |  |  |
| Less than 1 year | 3 | 0 | 3 |
| 1–5 years | 3 | 7 | 10 |
| Total | 6 | 7 | 13 |

The findings are presented by research question.
NGOs, non-governmental organisations.

**Table 2** Description of the documents included in the desk review for Kenya and Tanzania

| Type of document | Name of document | Country | Dates | Content description |
|---|---|---|---|---|
| Policy[20] | Tanzania Health Policy 2007 | Tanzania | 2007 | Guides development of NCD guidelines, emphasising health service provision, equity, capacity building and awareness on NCD management and prevention. |
| Policy[21] | Kenya Health Policy 2014–2030 | Kenya | 2014 | Provides overall direction for significant health improvement, aligning with Kenya's long-term development agenda and global commitments to address the rising burden of NCDs. |
| National Framework[22] | National Strategy for the Prevention and Control of NCDs 2015–2020 | Kenya | 2015–2020 | Emphasises an integrated approach to reduce the burden of NCDs, providing evidence-based interventions for sustainable socioeconomic development. |
| National Framework[23] | Health Sector Strategic Plan (HSSP IV) | Tanzania | 2015–2020 | Advocates for integrating NCD clinics into the healthcare system, emphasising access to care, NCD prevention and early detection. |
| National framework[24] | Strategic and Action Plan for the Prevention and Control of NCDs in Tanzania 2016–2020 | Tanzania | 2016–2020 | Advocates for NCD prevention and control at the national level, emphasising leadership, governance, multisectoral collaboration and accountability. |
| Guideline[33] | National Guideline of Clinical Management and Infection Prevention and Control of Novel Coronavirus (COVID-19) | Tanzania | April 2020 | Educates on COVID-19 preventive measures, emphasising accurate information provision for people with NCDs. |
| Guideline[28] Guideline[29] | Interim Guidelines on Management of COVID-19 in Kenya Updated version: Guidelines on Case Management of COVID-19 in Kenya | Kenya | April 2020 July, 2021 | Combines preventive and clinical management guidelines for COVID-19, updated version includes the latest clinical management guidelines. |
| Letter[35] | Non-Communicable Diseases Clinic During COVID-19 Outbreak | Kenya | Mid-April 2020 | National directive for NCD clinics to remain operational during the COVID-19 pandemic. |
| Guideline[34] | Guidance on the Provision of NCD and Mental Health Services in the Context of the COVID-19 Outbreak in Tanzania | Tanzania | May 2020 | Targets strengthening NCD service provision during COVID-19, including capacity building, health promotion, facility restructuring and continuity of essential services. |
| Guideline[30] and Guidelines[31] | Interim Guidance on Continuity of Essential Health Services during the COVID-19 Pandemic. Updated version: Guidance on Continuity of Essential Health Services during the COVID-19 Pandemic | Kenya | May 2020 July 2020 | Provides guidance for reorganising the health system to ensure continuity of health services during the COVID-19 pandemic, including NCD services. |
| Guideline[32] | Interim Guidance on Provision of Services for NCDs during the COVID-19 Pandemic | Kenya | July 2020 | Highlights COVID-19-related challenges on NCDs and their management. |
| Report[36] | Flash appeal for COVID-19 Tanzania | Tanzania | July–December 2020 | Discusses capacity building for health workers, mentorship on NCD and COVID-19 management, information dissemination and health promotion strategies. |
| Policy Brief[25] | Tanzania NCD Alliance (TANCDA) policy brief on the inclusion of NCDs on universal health coverage in Tanzania | Tanzania | March 2021 | Recommends actions for quality NCD care, health financing, multisectoral collaboration and health awareness during the COVID-19 pandemic. |

**Table 2** Continued

| Type of document | Name of document | Country | Dates | Content description |
|---|---|---|---|---|
| Policy[27] | Kenya Emergency Medical Care Policy 2020–2030 | Kenya | July 2021 | Provides guidance, structures, resources, regulations and standards for establishing an emergency medical care system, addressing emergency conditions caused by NCDs. |
| National framework[26] | National Strategic Plan for the Prevention and Control of Non-Communicable Disease 2021/2022–2025/2026 | Kenya | August 2021 | Provides direction on sectoral and multisectoral coordination, leadership and governance for NCD prevention, research promotion and surveillance enhancement. |

NCDs, non-communicable disease.

Health Policy 2007,[20] the Kenya Health Policy 2014,[21] the Kenya NCD Strategic Plan 2015–2020,[22] the Tanzania Health Sector Strategic Plan (HSSP IV) 2015–2020[23] and the Tanzania NCD Strategic Plan 2016–2020.[24] The Tanzania Health Policy 2007[20] aims to provide guidance about provision of health services for people with NCDs, but does not include any advice regarding emergency preparedness actions in relation to NCDs. The Kenya Health Policy[21] provides general guidance on health system response to national disasters, emergencies and disease outbreaks, however, it does not provide specific guidance in relation to NCD management. The Health Sector Strategic Plan in Tanzania[23] reinforced the importance of integration of NCD clinics into the healthcare system to enhance accessibility of care, but emergency care for NCD was not addressed. The Kenya NCD Strategic Plan 2015–2020[22] and Tanzania NCD Strategic Plan 2016–2020[24] addressed NCDs. Both documents promote preventive, curative and rehabilitative services and advocate strengthening leadership and governance, and multisectoral collaboration with an emphasis on accountability for the prevention and control of NCDs. However, these strategic plans do not refer to NCD care during emergency situations.

In both countries, participants identified policies, national frameworks, and guidelines for managing NCDs including T2D, which corroborates the findings from the review. KIs also noted that the documents did not cover aspects of care during a pandemic/emergency situation.

Kenya didn't have guidelines that govern diabetes and COVID-19 (pandemics or emergencies) but there were general guidelines for the management of both type two and type one diabetes. (Policy Actor_1905_KE)

### Modified documents

There were two documents; one from each country that had been modified during the pandemic to adjust for changing circumstances. In both countries, the NCD strategic plan documents were expiring hence the pandemic

situation created an opportunity to highlight emergency situations in the lapsing NCD strategic plans.[25 26] The pre-existing NCD strategic plans[22 24] were modified to include NCD care during the COVID-19 pandemic. KIs confirmed that there were gaps in the previous NCD strategic plans. They described that the COVID-19 pandemic caught them unprepared as there were no plans for continuity of care for NCD patients during this period and this necessitated the modification of the NCD strategic plans during the COVID-19 period.

…Before the onset of COVID-19, I would want to say that we did not have policies or guidelines to manage diabetes (during emergencies) and therefore when COVID-19 came it presented us with some challenge. It was also an eye opener as far as the management of the NCDs is concerned during emergencies or during when we have natural disasters. (Policy Actor_1511_KE)

We had the strategic plan for the years 2016 until 2020, which itself was inclusive, so when COVID-19 occurred, we found that since it affects patients living with non-communicable diseases. We should only develop a guideline that will be different from the previous one, because the previous one was on leadership and awareness matters but it (NCD strategic plan 2016–2020) had not described the COVID-19 pandemic. (Policy Actor_0002 _TZ)

### New documents developed during the pandemic

We identified 10 new documents that were developed during the pandemic to manage NCD patients: seven from Kenya and three from Tanzania. These included: one policy (Kenya)[27] and seven guidelines (five in Kenya[28–32] and two in Tanzania[33 34]) and two other documents which include a government directive in Kenya[35] and a report in Tanzania.[36] All these documents were created to provide specific guidance on the provision of essential health services and clinical management of NCDs during the pandemic. These documents also provided NCD patients with information on how to manage NCDs during the pandemic.

### Research question 2: what do the national frameworks, policy models or guidelines include for management of type 2 diabetes/NCDs in pandemic/emergency situations?

In both countries, most of the documents reviewed included aspects of national priority setting for managing NCDs in pandemic/emergency situations. The subsequent sub-sections highlight the content of the documents.

#### Content of the documents

In Kenya, the first interim guidelines for COVID-19 case management were released in April 2020.[28] The guide provided advice on both preventive and clinical management of COVID-19 in Kenya. The interim guidelines were adapted from various international recommendations including from the WHO. An updated version of the guideline was later released in July 2021,[29] and it contains the latest guidelines for the clinical management of COVID-19 in Kenya.

Another interim guideline for essential health services during COVID-19 was released in May 2020.[30] This document was targeted towards healthcare managers and healthcare workers to ensure that people continued to receive essential health services. This was followed by the release of an updated version of the guideline in July 2020.[31] The latter provided guidance for healthcare providers on immediate actions that should be considered to reorganise the health system to ensure continuity of health services during the COVID-19 pandemic for all services including NCD services.

Also in July 2020, another interim guidance on the provision of services from NCDs during the COVID-19 pandemic[32] was released. This document highlighted specific COVID-19-related challenges on NCDs including severity among high-risk persons, COVID-19 actions/responses jeopardising access to care and disrupting lifestyle approaches, disruption of funding and supplies and the general management of NCDs.

A letter from the Director General of Health in Kenya[35] to the County Executive Committee members for health was also among the first few documents released in mid-April 2020 to request continuity of NCD care among the disruption to NCD care provision during the COVID-19 outbreak. This directive instructed NCD clinics to remain operational in all counties during the COVID-19 pandemic.

In addition, a new Emergency Care Policy,[27] distinct from the preceding guidance, was issued in July 2021 to provide a roadmap for emergency services to all who need it including those with NCDs. Finally, in August 2021, the new Kenya National NCD Strategic Plan[26] was launched; it highlighted the disruption to healthcare caused by the COVID-19 pandemic and the need for resilient health systems during health emergencies. It further describes the NCD strategic plan being sensitive to public health emergencies, such as the COVID-19 pandemic but it does not provide guidance on how to better prepare for future emergencies.

In Tanzania, two documents were developed in the early phase of the pandemic. The first was a set of guidelines focused on educating the public on COVID-19 prevention measures, released in April 2020, and it emphasised the need to provide accurate information about the COVID-19 pandemic for patients with NCD.[33] In May 2020, the MOH in Tanzania released another guideline on the provision of NCD and mental health services in the context of the COVID-19 outbreak.[34] This guideline sets out plans for maintaining access to quality healthcare for NCDs and mental health services during and beyond the pandemic and for strengthening access to COVID-19 pandemic education. It emphasises the importance of protecting people with chronic NCDs, such as T2D patients, from COVID-19 and providing recommendations for doing so, including hand hygiene, maintaining glycaemic control, staying in touch with healthcare providers, refilling medications before running out and going to health facilities if experiencing COVID-19 symptoms. Additionally, it provided healthcare providers in Tanzania with information on how to manage COVID-19 infection and comorbidities including the major NCDs.

Between July and December 2020, a report on COVID-19 was released in Tanzania to strengthen the capacity of health workers to manage NCDs and COVID-19.[36] In March 2021, the MOH in Tanzania in collaboration with the Tanzania NCD Association (TANCDA) issued a policy brief[25] instead of updating the NCD strategic plan[24] to inform the care aspects of NCDs including diabetes during a pandemic situation. This policy brief reinforced the importance of improving diabetes and NCD management during COVID-19 by empowering NCD patients to take charge of their health, engaging different multisectoral groups to support NCDs health services financially and to promote health awareness.

#### Access to care and medical supplies

The documents reviewed described various approaches to care such as modifications to regular practice, workforce issues, how to respond to care disruptions and health education. In Kenya, several documents were developed during the pandemic to minimise disruptions in care for patients with NCDs due to COVID-19.[30–32] These documents provided mitigation measures such as provision of adequate supply of drugs to patients, patient virtual support (through telemedicine, the use of community health volunteers (CHVs) and helplines), counselling, provision of personal protective equipment for healthcare providers and maximum social distancing during facility visits. The Kenya Health Policy[21] provides the long-term development agenda[37] and global commitments. One of its objectives is to halt and reverse the rising burden of NCD. Access to care and supplies of medicines are discussed, but it is not specific to NCDs. The letter by the Director General for Health to all NCD clinics in Kenya noted that access to NCD care was hampered by the closure of the NCD clinics early into the pandemic.[35] This directive addressed the NCD access

issue during the pandemic by instructing NCD clinics to remain operational in all counties during the COVID-19 pandemic. The interim guidance on the provision of services for NCDs during the COVID-19 pandemic[32] provided specific information to ensure care continuity for key NCDs by informing patients of their increased risk of COVID-19, supporting patients to self-manage when appropriate, increasing patients' home supplies of medication and stock needed for their monitoring devices, providing counselling on healthy diets and physical activity, ensuring maximum social distancing during their visits, modifying their routine clinical review frequency and means/mode of delivery. This guideline further provided a section on how to manage diabetes during the COVID-19 pandemic. The document provided advice on the strategies that people with diabetes can adopt to reduce their risk of COVID-19. This document further acknowledged that there may be instances when the limited healthcare resources could be diverted to care only for those with COVID-19. Some of the documents reviewed encouraged the adoption of virtual consultations and digital diagnostic platforms.[26 28 31 32]

In Tanzania, the guideline[34] and report[36] recognise that the continuity of NCD care was important because of how vulnerable people living with NCDs were during the pandemic. The guideline also highlighted the importance of accessing care to minimise the disruptions in the healthcare system. The guideline emphasised patients' access to essential services during the pandemic, and it recommended free provision of all preventive care medical supplies.[34] The report also advocated for increased provision of essential medical supplies and access to essential diagnostic equipment and treatment for NCDs during the pandemic.[36]

The KIs in both countries described how access to care was hindered during the pandemic for several reasons including patients with NCDs not wanting to go to health facilities out of fear of contracting the virus and limited skills of healthcare workers to handle NCDs and COVID-19. KIs also confirmed that policies and guidelines were developed to address the access issues experienced during the COVID-19 pandemic.

> You know, most of the health facilities were shut down, generally the whole country was shut down including health facilities and so what remained were just facilities to respond to the COVID. Because of that, several of our clients could not access the services that they required. A letter was sent to the counties by the acting Director General for health to have these clinics open. So that was one of the policies that was issued so that the clients would be able to access the services. (Policy Actor_1605_KE)

> NCD services were affected in some clinics. Service providers lacked skills, so they didn't know what to do to serve NCD patients. In some facilities, some providers stopped providing services and some clinics were closed, and the patients were afraid to go to the

clinics. As a result, it was a challenge on patients' and providers' sides. (Policy Actor_0002_TZ

KIs also described that continuity of care for NCD patients was done using alternative strategies. Healthcare providers were encouraged to ensure continuity of care by following up with their NCD patients at the community level either physically with the help of CHVs or virtually.

> …We encouraged the clinicians to do close monitoring and contact tracing with the help of the community volunteers and in facilities which were more advanced, they did it (followed up with patients) virtually. (Policy Actor_0912_KE

### Monitoring and evaluation tools

There were three documents that provided monitoring and evaluation frameworks; the Kenya Health Policy,[21] the National Strategic Plan for the Prevention and Control of Non-Communicable Disease 2021/2022–2025/2026[26] and the Kenya Emergency Medical Care Policy 2020–2030.[27] The Kenya Health Policy includes progress indicators across eight domains and their targets are based on the WHO statistics of the average value of four middle-income countries. One of the targets is to reduce annual deaths due to NCDs by 27% in 2030. Some of the targets included in the National Strategic Plan for the Prevention and Control of NCD 2021/2022–2025/2026[26] are to increase the proportion of funding to NCDs from 48% to 60%, reducing the prevalence of high blood pressure by 25% and reducing the rising burden of diabetes by 10% in 2025. The Kenya Emergency Medical Care Policy 2020–2030[27] monitoring and evaluation framework had general targets for emergencies with nothing specific to NCDs. Even though several documents reviewed in Kenya had monitoring and evaluation frameworks, none were focused on emergency/pandemic care. In Tanzania, none of the documents reviewed mentioned monitoring and evaluation tools.

### Research question 3: what are the policy/guideline gaps for the management of type 2 diabetes during the COVID-19 pandemic, and what are the opportunities to improve the management of T2D/NCDs?

#### Gaps identified

Several gaps were identified from the reviewed documents and the KIIs from the two countries. These included limited resource allocation and the lack of comprehensiveness of the documents with future preparedness and monitoring and evaluation tools inadequately addressed.

#### Resource allocation

In both countries, resource allocation for the management of NCDs in pandemic/emergency situations was not described in the guidance/policy documents. KIs noted that this affected the development, dissemination and implementation of the new policies/guidelines.

KIs stated that the development of the new policies/guidelines was affected by the COVID-19 restrictions on

gatherings. In the initial stages of COVID-19, all meetings for the policy/guideline development were moved to virtual platforms. Participants said that the technology needed to access virtual meetings (eg, data bundles, modems) was under-resourced, making participation in the policy/guideline development meetings challenging and strenuous. KIs also suggested that the virtual policy/guideline meetings struggled to reach consensus.

Yes, there were challenges (…) The guideline was done quickly but it was still a tug of war in people's participation: sometimes other people delayed in responding, others did not submit their input, or sometimes others submitted their input after the document has already moved further steps, sometimes a person could participate today or tomorrow, and he does appear (not participate again). (Policy Actor_0003_TZ)

We had no resources, so we bought our modems. We had to ensure that the (data) bundles were always enough because the meetings were over and over through virtual meetings, so like every other minute you had a virtual meeting. There was no direct resource to the NCD department for COVID-19. (Policy Actor_1905_KE)

Dissemination of the new policies/guidelines was challenging due to inadequate resources to cover all regions and the COVID-19 restrictions. Participants mentioned that not all the healthcare workers received copies of the new guidelines. Others mentioned that many healthcare workers were not trained on the new guidelines.

I think the resources were available but probably not adequate. So, for instance, the financial resources to support a full-fledged dissemination across the country…or to support mentorship across all the counties was not adequate. (Policy Actor_1401_KE)

About the dissemination, you will remember during the COVID period, the matter of calling people together was restricted, so the dissemination phase was not done properly although I know there was training. For example, TANCDA trained people in Dar es salaam and the Ministry of Health was also doing training in some regions… It was just the distribution of the guideline (was inadequate). We just shared this document on the online platform. (Policy Actor_0003_TZ

In both countries, the implementation of the new guidelines was a challenge. Participants described the monitoring and evaluation of the new guidelines as a gap due to the limited resources allocated and the restricted movements due to COVID-19. KIs mentioned that they were not aware of how the implementation was progressing.

So, I may not be able to speak about implementation because I'm not sure what happened after the guidelines were released and shared widely through emails and WhatsApp groups. Whether the healthcare workers used the guidelines or whether they were important or whether they were asking for a revision, or the guidelines were speaking to them, I am not able to say anything about that. I don't know because we had no resources to monitor the implementation. (Policy Actor_1905_KE)

…since there was no follow-up done, we were not sure whether the printed 500 copies of the guideline reached the respective health facilities (Policy Actor_0002_TZ

The reallocation of most healthcare funding to address COVID-19 meant that there were no funds to cater for the recommendation to give patients with T2D a longer supply of NCD medicines as referred to in reviewed documents in Kenya.[30–32] Interviews with the policy stakeholders in Kenya and Tanzania further confirmed this. In Tanzania specifically, the health insurance coverage was not adjusted to comply with the new guidelines; therefore, the implementation of the recommendation to give patients with T2D a 2–3 month supply of medications to limit hospital visits was hampered.

Other support resources are required, (for example,) if someone is to treat a patient … then you need those medicines in place. (However,) the main availability of NCD medicines in the country is half of what is required; it's like 42% out of the targeted 80%…. therefore, there is an inadequate amount to enhance full implementation of these guidelines. (Policy Actor_1401_KE)

I remember at one point we even asked the hospitals not to give longer provision of supplies … if for example, you give them three months' supplies it means that you are denying someone else, so we decided to go back to the one month supply of medicines and then monitor the situation. (Policy Actor_1905_KE

## Comprehensiveness of the new policies/guidelines

A gap identified in the review of the documents was the lack of tools needed to monitor and evaluate the progress of the new policy/guidelines. In both countries, KIs noted that the guidelines developed lacked monitoring and evaluation tools.

Sometimes we don't embed (monitoring and evaluation tools) in the supervision (of health facilities), instead, they are included in the supervision tool. Okay, so these guidelines that are disease-specific or aimed at patient management, we don't include them with the M&E framework unless we have a separate framework for supervision. (Policy Actor_0003_TZ)

None of the policy/guidelines in Tanzania included plans for managing T2D/NCDs in emergency situations. Participants described this to be a gap even in the

new policy/guidelines that were developed during the COVID-19 pandemic.

> Another thing we have come to discover later is that the guideline was still not enough to provide guidance on what to do in an emergency to care for patients with NCDs. (Policy Actor_0001_TZ)

### Opportunities identified

Participants were able to provide insights into what opportunities they took advantage of to address some of the challenges. KIs in both countries mentioned that the pandemic created an opportunity to establish online platforms to ensure information reached key people within the healthcare sector.

> …in terms of monitoring or supportive supervision that happened maybe not to the extent that we want… but one of the ways we are collecting feedback is (by) empowering the counties and having an NCD focal person… Now we have a common platform both through email (and) we have a WhatsApp group for them where we quickly disseminate any required information… (Policy Actor _1004_KE)

In Tanzania, the COVID-19-pandemic presented an opportunity to use routine supervision meetings as an avenue to disseminate the new policies/guidelines to health facilities. The intention was that the information would be cascaded downwards to all health facility staff.

> …but we have also been able to disseminate in the sense of going with the guidelines when we do supervision, this was done in the first group that I was involved in, when we meet the regional health management team (RHMT) we take them through the NCD guideline, but we also provide them with copies for them to distribute in the respective health facilities, we did not go to all council health management team (CHMTs). (Policy Actor_0001_TZ

Participants also mentioned the importance of putting forth strategies to strengthen the health system response towards managing NCDs including diabetes in general and during emergency situations. This included prioritising NCDs, multisectoral action, multilevel engagements and improved data and monitoring systems to ensure continuity of care.

> … (There) should be a multidisciplinary preparedness team which should keep on meeting frequently and if need-be on monthly basis. (…) We (also) need to have a very elaborate policy statement as far as management of NCDs during a pandemics is concerned, and this policy should talk of resources strengthening of our systems, monitoring and evaluation, surveillance, and research. (It should also be able to strengthen the capacity of our healthcare providers.… It should also address the issue of medicines and health technologies which are available such

that even if the patient does not go to a health facility they can seek service from where one is… (Policy Actor_1511_KE)

## DISCUSSION

This study provides a comprehensive review of policies/guidelines that exist to manage NCDs in Kenya and Tanzania during emergency situations with inputs of key NCD stakeholders involved in decision making. This review revealed that prior to the COVID-19 pandemic, both countries did not have comprehensive policies/guidelines that addressed the management of NCDs during pandemic/emergency situations. However, several measures were put in place during the COVID-19 period to address the needs of patients with NCD and T2DM. These measures included the following: updating and developing strategic plans; development of new care guidelines; and release of government directives that guided continuity of care strategies. However, insufficient attention was given to implementation of the new policies and guidelines in both countries. There was inadequate guidance and resources provided to enable effective monitoring and evaluation of progress in relation to disaster preparedness.

The need to develop new and update policies/guidelines in Kenya and Tanzania was similar to other countries.[10] Our study found that the implementation of new policies/guidelines in Kenya and Tanzania faced challenges, for example, inadequate resources hampered the dissemination and training of healthcare workers. There were a lack of key performance targets and a lack of monitoring and evaluation which resulted in poor implementation and reduced effectiveness of the guidelines. A systematic review has revealed that NCD policy development and implementation is often inadequate in Africa.[38] A possible explanation is that healthcare resources are already limited and during the pandemic, these resources were diverted to address COVID-19 infections.[15] Research conducted by the International Federation of Red Cross (IFRC)[39] during the COVID-19 pandemic looking at the role of law and policy in public health emergency preparedness and response found that the pre-existing policies and laws were inadequate to cater to the current COVID-19 pandemic. This was similar to the finding of the study reported here. Due to the rapid development of the new policies/guidelines, a weakness noted was that major components were missed such as monitoring and evaluation and emergency preparedness tools in most of the policies/guidelines.

Some findings were similar across the two countries. For instance, in the initial phase of the COVID-19 pandemic, KIs reported disruption to healthcare services for patients with NCDs in both countries. Therefore, access, availability and utilisation of NCD care services were severely affected. The disruptions noted in Kenya and Tanzania were similar to disruptions reported across multiple

countries;[3][10] for example, a WHO report said 75% of countries experienced disruptions to various NCD care services.[10] Supply of NCD medicines was a major issue at multiple levels of the health system,[40] and many people living with NCDs lacked sufficient medicine supplies to manage their conditions, therefore leading to worse health outcomes. There was further evidence to show that LMICs especially countries in SSA were more likely than other countries to report disruption of care due to unavailability/stock-outs of medicines and reduced number of healthcare providers at heath facilities[10][41] as was reported by KIs in the current study. The most common reasons reported for not accessing care among people living with NCD in a recent study conducted by Devi *et al* were fear of contracting COVID-19 (36.5%), doctors' advice not to go to the health facility (33.7%) and physical inaccessibility due to the lockdowns (27%).[42] This meant that many patients with chronic diseases were not able to receive the care that they needed and resonates with the findings presented here.

The current study shows that both countries had gaps between what was stated in the policy and what was occurring in practice and this was similar to findings from a previous study conducted before COVID-19 looking at the development and implementation process of NCD prevention policies in SSA.[43] Despite the newly developed or updated policies/guidelines highlighting the need for longer supply of medicines for NCDs, both countries were unable to fully comply with these policies/guidelines due to low medication stocks. In addition, healthcare systems in Tanzania failed to comply with the new policies/guidelines because health insurance schemes were not aligning themselves with the new policies/guidelines.

### Strengths and limitations

A major strength in this study was the triangulation of data sources from the desk review with qualitative interviews. Another strength is the use of a comprehensive approach to identify relevant documents and extracting data in a standardised format. Key stakeholders involved in NCD decision-making further assisted us in retrieving documents not in the public domain. However, the findings from this study are limited to information from documents that were documented and accessible for review. Another limitation of this review is that we did not include patients 'and healthcare providers' perspectives, which would have been valuable in understanding the implementation process of the new/updated policies/ guidelines as well as the successes and challenges in order to inform recommendations for improvement.

### Recommendations

The COVID-19 pandemic created an opportunity for governments to review and update their national disaster plans. Existing government national disaster plans should integrate NCD management guidelines with national disaster preparedness plans. The plans should also have adequate ring-fenced allocation of resources to ensure

proper implementation and monitoring and evaluation of effectiveness of guidelines during emergencies and future pandemics. Governments should strengthen health systems and emergency preparedness by ensuring continuity of service provision for patients with NCDs by offering innovative services such as online consultation platforms, home delivery of diagnostics and ensuring adequate supplies of medication. Additional research on this topic has the potential to offer valuable insights on future developments.

### CONCLUSION

By comprehensively analysing the policy response to managing chronic diseases in Kenya and Tanzania, we have identified the effect of the pandemic on continuity of care. Kenya and Tanzania have developed and updated new policies/guidelines to include an emphasis on continuity of care for people with NCDs. There is a pressing need to develop disaster/emergency preparedness plans that encompass NCD care and allocate sufficient resources to ensure their effective implementation. This is crucial for effectively managing future emergency situations while strengthening the resilience of health systems to withstand multiple shocks. There is also need to promote and build capacities for implementation research in Kenya and Tanzania.

**Author affiliations**
[1]Chronic Disease Management Unit, African Population and Health Research Center, Nairobi, Kenya
[2]Department of Health System, Impact Evaluation and Policy, Ifakara Health Institute, Dar es Salaam, Tanzania
[3]School of Health and Wellbeing, University of Glasgow, Glasgow, UK
[4]Wisdom Consulting, New York, New York, USA
[5]School of Social and Political Sciences, University of Glasgow, Glasgow, UK

**Acknowledgements** We thank the field staff in both countries for their dedication and commitment. We also extend our thanks to the experts who generously shared their time, knowledge and experiences, which were fundamental to the success of this study.

**Contributors** SFM, LK and IM extracted and synthesised data from the policies and analysed the qualitative data from both countries. SFM and LK jointly conducted the literature review and analysis and wrote the first draft manuscript. All the other authors (IM, FM, JPW, CB, CG, PMK, RES, CHK, SMM, PB, and GA) reviewed the draft manuscript, provided input, provided critical comments, and suggested additional analyses. All authors contributed substantially to the development, revision, and finalisation of the manuscript. SFM finalised the manuscript, which was subsequently approved by all authors. SFM is the guarantor.

**Funding** This work was supported by the Medical Research Council and by the National Institute for Health Research (NIHR) (grant number MR/V035924/1) using UK aid from the UK Government to support global health research. The views expressed in this publication are those of the author(s) and not necessarily those of the MRC, NIHR or the UK government.

**Competing interests** None declared.

**Patient and public involvement** Patients and/or the public were not involved in the design, or conduct, or reporting or dissemination plans of this research.

**Patient consent for publication** Consent obtained directly from patient(s).

**Ethics approval** This study involves human participants and was approved by Ethical approval was granted by the Ethics and Scientific Review Committee at AMREF Health Africa in Kenya (ESRC P900-2020) and the National Institute for Medical Research (NIMR/HQ/R.8a/Vol.IX/3806) in Tanzania. Participants gave informed consent to participate in the study before taking part.

**Provenance and peer review** Not commissioned; externally peer reviewed.

**Data availability statement** Data are available upon reasonable request. Only the qualitative data can be made available upon reasonable request. All other relevant data are available publicly.

**ORCID iDs**
Shukri F Mohamed http://orcid.org/0000-0002-8693-1943
Lyagamula Kisia http://orcid.org/0000-0002-2045-6158
Frances Mair http://orcid.org/0000-0001-9780-1135
Cindy Gray http://orcid.org/0000-0002-4295-6110
Richard E Sanya http://orcid.org/0000-0001-6348-9075
Caroline H Karugu http://orcid.org/0000-0002-6914-9714
Peter Binyaruka http://orcid.org/0000-0002-1892-7985

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
