## [Reviewer comments · BMJ Open]

ARTICLE DETAILS

TITLE (PROVISIONAL)	Policies for type 2 diabetes and non-communicable disease management during the COVID-19 pandemic in Kenya and Tanzania: a desk review and views of decision makers
AUTHORS	Mohamed, Shukri; Kisia, Lyagamula; Mashiashi, Irene; Mair, Frances; Wisdom, Jennifer P.; Bunn, Christopher; Gray, Cindy; Kibe, Peter M.; Sanya, Richard; Karugu, Caroline; Mtenga, Sally M.; Binyaruka, Peter; Asiki, G

VERSION 1 – REVIEW

REVIEWER	Hacker, Karen A. Centers for Disease Control and Prevention
REVIEW RETURNED	23-May-2023

GENERAL COMMENTS	This is a timely and helpful article that demonstrates the lack of attention to non communicable disease during the COVID-19 pandemic. While this focused on two African nations, I assume it was similar circumstances in other countries as well. The methods are well defined and the addition of qualitative interviews is very helpful. As noted in the limitation, it would have been very helpful to obtain the view of some providers. I wonder how they dealt with their patients with NCD during the pandemic and whether they considered their own strategies to be successful. It would be helpful to note that more research in this area would be illuminating for the future.
--

REVIEWER	Barghi Shirazi, Fahimeh Iran University of Medical Sciences
REVIEW RETURNED	11-Jun-2023

GENERAL COMMENTS	Greetings and Regards The manuscript is very good, but it needs minor corrections, if the respected author and Ekitor deem it appropriate, the corrections can be made Thanks (The reviewer provided a marked copy with additional comments. Please contact the publisher for full details.)
---

REVIEWER	Ben Mansour, Nadia Cardiovascular Epidemiology and Prevention Research Laboratory, Faculty of medicine of Tunis
REVIEW RETURNED	01-Sep-2023

GENERAL COMMENTS	The article describes an original and relevant desk review of NCDs management policies/guidelines during crisis in Kenya and
--

	Tanzania, before and during COVID 19 epidemic. Another important feature of this work are inputs of key NCD stakeholders involved in decision making. This review revealed that prior to the COVID-19 pandemic, both countries did not have comprehensive policies/guidelines that addressed the management of NCDs during pandemic/emergency situations. In spite of the strategies developed during the epidemic to address NCD and T2DM continuity of care, insufficient attention and resources were given to their implementation. It is evident that these results are consolidating the available existing body of research on NCDs management resiliency during crisis and health system analysis in general, which are very limited in African countries. However, some methodological details as well as specific public health implications based on the evidence generated have to be addressed. Recommendation: revise and resubmit Minor revision title: the research question and the results are focusing on T2DM and NCDs, it would be better to use these terms in the title, especially given that “chronic disease” was not used in the manuscript. Page 3 “What do the new findings imply” The two Recommendations are overlapping, they can be reformulated into one. Promoting and building the capacities for implementation research in both countries can be a relevant recommendation Introduction: 1) Authors should present a brief overview assessing the toll of the epidemic in the introduction to further clarify the context and the pressure on the policy decision makers. 2) Authors should update the references in the introduction, more references must be provided. Methods: 1) Selection criteria of the key informants were not clarified 2) Ethical considerations: Even though patients were not involved in the data collection, an IRB approval must be provided since interviews were conducted with key informants. Discussion: 1) Page 23 line 8: “in adequate” had to be corrected 2) Patients’ perspectives were not studied, that should be also stated as a limit of the study. 3) Public health implications, seem too “generic”, less specific to both countries and lack of practical aspects.
--	--

VERSION 1 – AUTHOR RESPONSE

Reviewer: 1: Dr. Karen A. Hacker, Centers for Disease Control and Prevention

Comments to the Author:

This is a timely and helpful article that demonstrates the lack of attention to non-communicable disease during the COVID-19 pandemic. While this focused on two African nations, I assume it was similar circumstances in other countries as well. The methods are well defined and the addition of

qualitative interviews is very helpful. As noted in the limitation, it would have been very helpful to obtain the view of some providers. I wonder how they dealt with their patients with NCD during the pandemic and whether they considered their own strategies to be successful. It would be helpful to note that more research in this area would be illuminating for the future.

Response: Thank you for this feedback and we appreciate. As much as we would have wanted to gather insights from healthcare providers regarding their experiences in managing patients with NCDs during the pandemic, unfortunately, we were not able to do and we acknowledge this as a limitation in our current study. We have also added the need for more research on this topic for the future in the recommendation section of the manuscript.

Reviewer: 2: Dr. Fahimeh Barghi Shirazi, Iran University of Medical Sciences

Comments to the Author:

*** Please find the comments from this reviewer in the attached file ***

Greetings and Regards

The manuscript is very good, but it needs minor corrections, if the respected author and Editor deem it appropriate, the corrections can be made

Thanks

Response: Thank you for your valuable feedback. We appreciate your careful review of this manuscript. We have made the necessary corrections as outlined below to respond to each of your comments.

Please explain more fully what are the entry and exit criteria?

Response: Thank you for this comment. Documents were included in this desk review if they met the following criteria under the Document review- eligibility criteria in the manuscript See page 6 under section titled "Eligibility criteria":

- Policy frameworks, policy statements, government statements/pronouncements, publications, emergency preparedness plans, strategies, laws, and operational guidelines for diabetes during COVID-19,
- Specifically mentions NCDs or diabetes or pandemic /emergency of COVID-19;
- Documents available in English and Kiswahili were included
- Documents that referenced Kenya or Tanzania

Documents were excluded from this review if they met any of the below criteria:

- The documents did not include information on managing type 2 diabetes in a pandemic/emergency situation in Kenya or Tanzania.
- The documents did not reference Kenya or Tanzania.

In the end, how many articles were extracted? Was your evaluation tool Czech?

Response: Figure 1 provides an overview of our search strategy, including the total number of documents identified and screened, the number excluded, and the number included in this review. Additionally, Appendix A outlines the search criteria employed for peer-reviewed literature.

It's important to note that this study involved a desk review, and in addition to this, we actively sought input from decision makers. Therefore, we did not employ an evaluation tool, as our approach was not that of a systematic review/meta-analysis.

How many articles were extracted? Was it valid and reliable?

Response: After applying the inclusion and exclusion criteria, a total of 17 policy/guidance documents were added to this review. This information is in the results section where we describe Figure 1 on Page 8-9, under the section titled "Results". The validity and reliability of the data extraction process were carefully maintained. All documents were catalogued in an online excel database which facilitated a standardized approach to data extraction. Additionally, the team held weekly "coding clinics" to ensure consistency in coding of the included documents and to ensure the accuracy and dependability of the extracted information. This information is on Page 6-7, under the "Search strategy" section.

Please send a sample search strategy to be reviewed.

Response: Thank you for requesting to review our sample search strategy. However, it's important to clarify that our approach was primarily focused on collecting and analysing policy documents, guidelines, and data related to non-communicable diseases (NCDs) with a specificity to type 2 diabetes rather than the development of a strategy. Our research primarily entailed reviewing and synthesizing existing policies and guidelines from a variety of reputable sources, as detailed in our methodology. Appendix A the specific terms used in the search for peer-reviewed literature and a detailed explanation of the search strategy for each data source of information is provided in Appendix B.

Please send the interview questions?

Response: The interview guide has been added to the supplementary materials as Appendix C.

Did you use a special software to analyze the interviews?

Response: We did not use any specialized software for the analysis of the interviews since we conducted a total of only 13 short interviews. As explained in the data management and analysis section Pg 7-8, the interview transcripts were manually coded in Microsoft Word. No special software was used to analyse the interviews as they were only 13 short interviews. As stated in the data management and analysis section, the interview transcripts were coded manually in word using a developed codebook.

How were the interviews analyzed, please explain?

Response: We used a content analysis method to analyse the information gathered from the interviews. This analysis involved an initial assessment of the nature of the documents in relation to priority setting. Subsequently, we identified gaps and opportunities within each of the reviewed documents. This information is provided on page 7-8 within the data management and analysis section.

Please limit the research?

Abbreviations?

Conflict of interest ?

thanking ? be mentioned at the end

Response: The above comments are not very clear but I have tried to respond to them. We have adhered to the journal specifications. All abbreviations are written out in full when first used. The

conflict of interest section was completed on the journal site. We also have completed the acknowledgement section. All these are either on the journal system or at the end of the article.

Reviewer: 3: Dr. Nadia Ben Mansour, Cardiovascular Epidemiology and Prevention Research Laboratory

Comments to the Author:

The article describes an original and relevant desk review of NCDs management policies/guidelines during crisis in Kenya and Tanzania, before and during COVID 19 epidemic. Another important feature of this work are inputs of key NCD stakeholders involved in decision making. This review revealed that prior to the COVID-19 pandemic, both countries did not have comprehensive policies/guidelines that addressed the management of NCDs during pandemic/emergency situations. In spite of the strategies developed during the epidemic to address NCD and T2DM continuity of care, insufficient attention and resources were given to their implementation.

It is evident that these results are consolidating the available existing body of research on NCDs management resiliency during crisis and health system analysis in general, which are very limited in African countries.

However, some methodological details as well as specific public health implications based on the evidence generated have to be addressed.

Recommendation: revise and resubmit

Response: Thank you for this comment.

Minor revision

Title:

The research question and the results are focusing on T2DM and NCDs, it would be better to use these terms in the title, especially given that "chronic disease" was not used in the manuscript.

Response: We appreciate your suggestion to use more specific terminology in the title. We have therefore revised the title to explicitly mention "T2DM" and "NCDs" to better reflect the content of the manuscript.

Page 3 "What do the new findings imply"

The two Recommendations are overlapping, they can be reformulated into one.

Promoting and building the capacities for implementation research in both countries can be a relevant recommendation.

Response: Thank you pointing out that two recommendations were overlapping. We have edited the two recommendations into one. We have also added your suggested recommendation which is crucial and we thank you for that. See page 3 under "What do the new findings imply?" for this revision.

- There is a pressing need to develop disaster/emergency preparedness plans that encompass NCD care and allocate sufficient resources to ensure their effective implementation. This is crucial for effectively managing future emergency situations while strengthening the resilience of health systems to withstand multiple shocks.
- There is need to promote and build capacities for implementation research in Kenya and Tanzania.

Introduction:

1) Authors should present a brief overview assessing the toll of the epidemic in the introduction to further clarify the context and the pressure on the policy decision makers.

Response: Thank you for this insight. An additional paragraph at the beginning of the introduction section has been included to provide a better context for the need for countries to respond to the magnitude of the pandemic.

2) Authors should update the references in the introduction, more references must be provided.

Response: Thank you for this comment. Additional references have been provided.

Methods:

1) Selection criteria of the key informants were not clarified

Response: Thank you for this comment. The selection criteria for key informants is on page 7 within the key informant interview section. "In both Kenya and Tanzania, key informants (KIs) were purposively selected based on their experiences and participation in NCD related policy formulation and implementation processes".

2) Ethical considerations: Even though patients were not involved in the data collection, an IRB approval must be provided since interviews were conducted with key informants.

Response: Thank you for this comment. IRB approval was sought for the interviews in both countries and the following ethics statement is within the research ethics approval section within the journal system. " Ethical approval was granted by the Ethics and Scientific Review Committee at AMREF Health Africa in Kenya (ESRC P900-2020) and the National Institute for Medical Research (NIMR/HQ/R.8a/Vol.IX/3806) in Tanzania."

Discussion:

1) Page 23 line 8: "in adequate" had to be corrected

Response: We have noted this spelling error and corrected it.

2) Patients' perspectives were not studied, that should be also stated as a limit of the study.

Response: Thank you for this comment. We acknowledge that one limitation of our study is the absence of patients' perspectives. We recognize the importance of including patient input in research to provide a comprehensive view of the topic. While this study focused on identifying the policy gaps in management of NCDs with a particular focus on diabetes during the COVID-19 period in Kenya and Tanzania, we understand that considering the perspectives and experiences of the patients would have enriched the depth of this research. We have acknowledged this in the manuscript as a limitation.

3) Public health implications, seem too "generic", less specific to both countries and lack of practical aspects.

Response: thank you for this feedback. We have edited this text in the conclusion section of the manuscript.

We thank the reviewers again for their thoughtful comments on the paper.

Shukri F. Mohamed – corresponding Author.